# Innate Lymphoid Cells and Their Role in the Immune Response to Infections

**DOI:** 10.3390/cells13040335

**Published:** 2024-02-13

**Authors:** Marek Fol, Wojciech Karpik, Agnieszka Zablotni, Jakub Kulesza, Ewelina Kulesza, Magdalena Godkowicz, Magdalena Druszczynska

**Affiliations:** 1Department of Immunology and Infectious Biology, Institute of Microbiology, Biotechnology and Immunology, Faculty of Biology and Environmental Protection, University of Lodz, 90-237 Lodz, Poland; marek.fol@biol.uni.lodz.pl (M.F.); wojciech.karpik@stud.umed.lodz.pl (W.K.); magdalena.godkowicz@edu.uni.lodz.pl (M.G.); 2Department of Bacterial Biology, Faculty of Biology and Environmental Protection, University of Lodz, 90-237 Lodz, Poland; agnieszka.zablotni@biol.uni.lodz.pl; 3Department of Internal Diseases and Clinical Pharmacology, Medical University of Lodz, 91-347 Lodz, Poland; jakub.kulesza@mp.pl; 4Department of Rheumatology and Internal Diseases, Medical University of Lodz, 90-549 Lodz, Poland; ewelina.kulesza@mp.pl; 5Lodz Institutes of the Polish Academy of Sciences, The Bio-Med-Chem Doctoral School, University of Lodz, 90-237 Lodz, Poland

**Keywords:** innate immune cells (ILC), intracellular pathogens, extracellular pathogens

## Abstract

Over the past decade, a group of lymphocyte-like cells called innate lymphoid cells (ILCs) has gained considerable attention due to their crucial role in regulating immunity and tissue homeostasis. ILCs, lacking antigen-specific receptors, are a group of functionally differentiated effector cells that act as tissue-resident sentinels against infections. Numerous studies have elucidated the characteristics of ILC subgroups, but the mechanisms controlling protective or pathological responses to pathogens still need to be better understood. This review summarizes the functions of ILCs in the immunology of infections caused by different intracellular and extracellular pathogens and discusses their possible therapeutic potential.

## 1. Introduction

Natural innate lymphoid cells (ILCs) are a newly described family of immune cells that form part of natural immunity [1,2]. They lack antigen-specific receptors, which allows them to act at the earliest stages of infection. According to the older nomenclature, lymphoid cells were grouped into ILC1s, ILC2s and ILC3s, which expressed the same functions as T helper 1 (Th1), Th2 and Th17 cells, respectively, with natural killer (NK) cells being the equivalent of cytotoxic T cells [3]. The currently proposed nomenclature classifies ILCs into five subgroups: NK cells, ILC1, ILC2, ILC3, and LTi (lymphoid tissue inducer) cells (Table 1). In addition, innate lymphoid regulatory cells (ILCregs) were identified and characterized due to the production of IL-10 and the absence of some of the markers described for ILC1: NK1.1, NKp46, and Tbx21 (encodes T-bet); ILC2: ST2, a lectin receptor-like killer cell subfamily G member 1 (KLRG1), and GATA-3; and ILC3: NKp46, CD4, and RORgt. The primary function of ILCregs is to maintain tissue homeostasis and regulate immune responses. They do this by secreting immunomodulatory cytokines, such as IL-10, TGF-β, and IL-35, which help in suppressing inflammatory responses and promoting tissue repair. ILCregs are found in various tissues, including the gut, lung, kidneys, and lymphoid organs. Each tissue may contain a distinct subset of rILCs with specific functions, tailored to the tissue microenvironment. Understanding their functions and dysregulation in various diseases may provide valuable insights for therapeutic interventions [4,5]. ILCs are distinguished by three main characteristics: (1) the absence of expression of lineage (Lin) markers, (2) the autonomy from recombination activating gene (Rag), and (3) the essential need for the common cytokine receptor γ-chain (γc) [6]. ILCs regulate immune responses, play a role in metabolic homeostasis, prevent tissue damage and tissue maintenance, and mediate responses between glial cells and other immune cells [7]. All ILC subpopulations, including NK cells, differentiate from a common lymphoid progenitor [8,9,10]. Different antigens have been proposed to identify each ILC group, but these have been described in detail elsewhere and are not the focus of this review [7,8].

## 2. Innate Lymphoid Cells (ILCs) Functional Diversity

ILC subpopulations vary considerably from tissue to tissue [11]. It is known that ILCs show great flexibility in cytokine production and are even capable of reversibly transforming into other subpopulations if necessary. ILC1s can transform into ILC3s and vice versa, ILC2s can transform into ILC1s, while NK cells have been shown to transform into ILC1s [12,13,14,15]. The understanding of ILCs’ function is still evolving. To date, ILCs have been described as cells mainly reflecting Th lymphocytes, but recently, cytotoxic subpopulations of ILC1 and ILC3 have been discovered [3,7,16,17,18,19]. ILC3s have shown high cytotoxicity and expression of granzyme B, granulysin and Eomes, previously thought to be expressed mainly in NK and CD8+ T cells [16,20]. ILC2 can also promote apoptosis through the production of PD-1 [21,22]. Another role that ILCs may play is in antigen presentation, as has been shown in specific populations of ILC3 and ILC2 expressing major histocompatibility complex (MHC) class II molecules [22,23]. This role is further supported by data obtained by Yang. et al., which show that the ILC2 subpopulation is capable of phagocytosis to some extent [24]. Recently, an interesting subpopulation of ‘regulatory’ ILCs in the small intestine was discovered, expressing constitutive IL-10 production in a mouse model, making them similar to Treg cells [25].

**Table 1 cells-13-00335-t001:** Characteristics of the subgroups of human innate immune cells.

ILC Subgroup	Phenotypical Markers	Transcription Factors	Cytokines	References
NK cells	CD56, NCR1, IL-12Rβ2	T-bet, Eomes	IFN-γ, TNF-α	[3,5,6,8,11,12,14,15,18,22]
ILC1	CD127,CD161, IL-1R, IL-12Rβ2, ICOS	T-bet, Eomes	IFN-γ, TNF-α	
ILC2	CD127, CD161, ICOS, CRTH2, IL-1R, ST2, IL-17RB	GATA3	IL-4, IL-5, IL-9, IL-13	
ILC3	CD117, CD127, CD161, NCR2, ICOS, NCR1, IL-1R, IL-23R	RORγt	IL-17, IL-22, GM-CSF	
LTi	CD117, CD127, IL-1R, IL-23R	TCF1	IL-2, IL-5, IL-13	

Abbreviations: CRTH2—chemoattractant receptor-homologous molecule expressed on Th2 cells; Eomes—T-box brain protein 2 (Tbr2); GATA3—GATA binding protein 3; ICOS—inducible T cell co-stimulator; IFN-γ—interferon-gamma; IL—interleukin; ILC—innate immune cells; LTi—lymphoid tissue inducer; NCR—natural cytotoxicity triggering receptor; NK—natural killer; RORγt—RAR-related orphan receptor γ; T-bet—T-box expressed in T cells; TCF1—T cell factor 1; TNF-α—tumor necrosis factor-alpha.

ILCs are lymphocytes that are resident in tissues, originating first in the fetal liver and subsequently in the bone marrow [7]. ILCs possess homing receptors such as CXCR5, CCR6, and CCR7, allowing some subpopulations to migrate to infected tissues [26,27,28], but the extent of this phenomenon is still debated. Migration rates can vary from a few percent to more than half of the ILCs recruited from circulation. This is highly dependent on the group and subpopulation of ILCs [28,29], and some infections may also lead to increased migration [26,30]. 

Despite the association of ILCs mainly with innate immune responses, they play an essential role in maintaining prolonged immunity to pathogens. Once activated, increased ILCs can persist for several months [31,32,33,34]. Moreover, all subgroups either mediate long-term memory responses [31,32,33,34,35,36] or exhibit memory-like properties. Naïve ILC2s have demonstrated a weaker response to unrelated allergens compared to post-inflammation persistent ILC2s [33]. This increase in the reactivity of innate immune cells is called ‘trained immunity’, and this phenomenon makes some standard vaccines a multipotent immune enhancer, resulting in an overall reduced mortality rate [37,38]. Another example of long-term immunity an ILC provides is the maintenance of CD4+ T-cell memory by LTi cells and ILC3 cells [32,35].

## 3. ILCs in Immunity to Intracellular and Extracellular Pathogens

Microorganisms are classified as extracellular pathogens that do not have to enter host cells to reproduce. They are present in their natural habitats as free-living entities, and they can also act as intracellular pathogens by infecting and reproducing within host cells via vacuolar or cytosolic routes. However, many intracellular bacterial pathogens can cause extracellular infection following the intracellular stage, and conversely, several extracellular pathogens can attack host cells in vivo prior to the extracellular infection stage [39]. Due to the wide distribution of heterogenous subtypes of ILCs across various tissues and organs in the body, these cells play a crucial role in the immune response against various pathogens, including viruses, bacteria, fungi, and intracellular and extracellular parasites (Figure 1 and Figure 2).

### 3.1. Mycobacterium tuberculosis (M.tb)

Due to their strategic location on the mucosal surface, ILCs can respond quickly to pathogen invasion. Through cytokine production and reciprocal interactions with macrophages, these cells have also been shown to be crucial in epithelial healing, tissue homeostasis, metabolism, and immune control. Although the presence of ILC cells in the lungs has been confirmed, their role, particularly in the course of tuberculosis, is poorly understood [40,41]. Among the three, ILC1 cells have been attributed a vital role in immune processes associated with infections caused by intracellular bacteria and parasites. Interestingly, although ILC1 and NK cells differ in development and transcriptional processes [42], they show some phenotypic similarities that allow them to be included in the same ILC1 group. In this context, attention is drawn to their similar reactivity to pro-inflammatory cytokines (interleukins (IL)-15, IL-12 and IL-18) and the secretion of IFN-γ and TNF as a manifestation of their activation [43]. The production of IFN-γ is regulated in ILC1 cells by the transcription factor T-bet, which is also present in NK cells. However, its expression level is lower in NK cells than in ILC1 and is not crucial for NK cell development [43].

Although the role of ILCs in *M.tb* infection has not been clearly defined, the data obtained so far leads us to conclude that they play a protective role, expressed mainly through the production of IL-17 and IL-22. In both mouse models and humans, it has been noted that TB infection leads to a significant reduction in the levels of ILC1s and ILC3s within the bloodstream, while these cell types tend to accumulate in the lungs. This process involves the participation of the CXCL13–CXCR5 axis, which is vital for coordinating interactions between B and T cells [44,45]. Using the murine model, Corral et al. demonstrated that during *M.tb* infection, the fate of individual ILC cell types varied: the number of ILC1 and ILC3 cells increased and became activated, while the ILC2 population shrank, and they were functionally inhibited [46]. It was discovered that an atypical population appeared in the lungs of infected mice that shared characteristics with ILC1s but were distinct, forming a group called “ILC1-like” cells. The inflammatory and metabolic environment was found to have influenced immature precursors of lung ILC cells, prompting them to differentiate into ILC1-like cells. The phenotypic and functional differentiation of lung ILC precursors expressing IL-18Rα into IFN-γ-producing ILC1-like cells was observed.

Moreover, using mice Rag2-/- deficient in T and B cells, the researchers showed that adaptive immune mechanisms were not required for the conversion of IL-18Rα+ILC to interferon-γ-producing ILC1-like population. The generation of ILC cells took place in a specific cytokine environment, namely a type 1 inflammatory environment, the development of which is observed during *M.tb* infection. The establishment of such an environment was favored by specific cytokines, such as IL-12 and IL-18, which induced IFN production by numerous immune cells (NK cells, T helper cells, and ILC1). Moreover, ILC1-like cells were potentially found to protect against *M.tb* infection, and interestingly, intranasal administration of the BCG vaccine provoked the production of these cells [46]. Furthermore, Tripathi et al. (2019) provided evidence in a mouse experiment where they showed that IL-22, produced by ILC3 cells, can enhance the survival of mice infected with *M.tb* and suffering from type 2 diabetes mellitus (T2DM). This effect is achieved by diminishing the inflammatory response, preventing the buildup of neutrophils around the alveoli, suppressing the production of neutrophil elastase 2 (ELA2), and safeguarding the integrity of epithelial cells. Thus, it is suggested that IL-22 may be a useful therapeutic tool in T2DM patients with tuberculosis infection, especially since serum levels of this cytokine are significantly lower in such patients relative to T2DM patients uninfected with the intracellular pathogen [47]. It was also observed that the number of ILC3s inducing Th17 response was lower in tuberculosis patients than in controls. An increase in the number of ILC1 and ILC3 was noted after treatment, suggesting that *M.tb* bacteremia affected selective ILCs accumulation. Pulmonary ILC3s accumulation was also associated with alveolar macrophage accumulation and increased *M.tb* control. In addition, the chemokine CXCL13, expressed in the lungs of mice and humans after *M.tb* infection, decreased in the plasma of patients with pulmonary TB after treatment. The expression of the CXCR5 receptor for CXCL13 was also increased in patients with active TB, suggesting that the CXCL13–CXCR5 axis may promote ILC3 activity after M.tb infection. Moreover, the addition of WT ILC3 infected *M.tb* restored early *M.tb* clearance in Rag2-/-IL2r-γ-/- mice, suggesting that harnessing the efficacy of ILC3s may be a novel pathway to induce TB control through Th17 immunity [26]. Reports suggest that Th17 immunity, which includes the effector cytokines IL-17, IL-22, and IL-23, might have a protective effect in *M.tb* infections, as shown in several animal studies [48,49,50,51]. However, IL-17 has also been implicated in causing tissue damage [52,53]. Pan et al. (2021) revealed that in the group of patients with active tuberculosis, the ILCs present in the circulation were upregulated and showed increased IL-17 production compared to the control group. A greater proportion of ILCs producing IL-17 (with IL-17 being the dominant secretion phenotype in ILC1) correlated with a more severe inflammatory state and a more unfavorable clinical condition [54]. Thus, determining the actual role of Th17-like ILCs remains a challenging task.

### 3.2. Salmonella Typhimurium 

*Salmonella* Typhimurium is the most commonly studied gastrointestinal pathogen within *Salmonella enterica* subspecies I, which typically causes infections after ingesting contaminated water or food [55,56]. These pathogens cause self-limiting inflammatory diseases of the gastrointestinal tract in humans, including acute inflammation of the terminal ileum and colon [57]. The bacteria must overcome several obstacles to reach the large intestine (the primary site of replication), including the colon. For example, they activate the acid tolerance response (ATR) and then use flagella and chemotaxis capacity to move close to the intestinal epithelium, consequently triggering an inflammatory response [55,58,59]. Interestingly, this process is not a typical host response leading to the elimination of the pathogen, but it is the result of pathogen-induced changes in the transcriptional processes of the infected cell (leading to the generation of exogenous electron acceptors), which in turn is thought to enable and sustain *Salmonella* Typhimurium replication [59]. This modification of host processes is enabled by activation of the Salmonella type III protein secretion system (T3SS). 

Increased susceptibility to *Salmonella* Typhimurium infection resulting from the depletion of NK cells (belonging to group 1 ILC) and loss of the ability to produce IFN-γ has been demonstrated in a mouse model, as well as using human peripheral blood-derived macrophages [60]. A protective role of IFN-γ against this enteric pathogen has also been confirmed in group 3 ILCs, and this antibacterial role has been linked to the absence of the transcription factor Runx3 [61,62]. Castleman et al. demonstrated that human colonic group 1 ILCs, when modeled in LPMCs, exhibit an elevated production of pro-inflammatory cytokines (specifically IFN-γ and TNF-α) upon exposure to *Salmonella* Typhimurium infection. Furthermore, the induction of IFN-γ production in these cells was found to be reliant on their exposure to IL-12p70, IL-18, and IL-1β [63]. In contrast, Kästele et al. confirmed increased levels of IFN-γ and some co-expression of GM-CSF in Rorγt+T-bet+ ILCs during the infection [64].

### 3.3. Severe Acute Respiratory Syndrome Coronavirus 2 (SARS-CoV-2) 

The SARS-CoV-2 coronavirus originated in China, and the COVID-19 (coronavirus disease of 2019) pandemic has been a significant global health concern in recent years [65]. It is known that the risk of severe COVID-19 and death is highly individual, and the course pattern can range from asymptomatic to severe multi-organ failure. The likelihood of experiencing severe illness resulting in hospitalization or death rises as individuals age, particularly in men. This risk is further heightened in individuals with comorbidities such as obesity and diabetes [66,67]. ILCs are commonly recognized as the initial defense mechanism against numerous infections, including viral ones [1,68,69]. To date, the role of ILC cells in respiratory-related infections has yet to be clearly defined. Although ILC2 cells predominate in the lungs, there is increasing evidence of a significant role played by ILC3s in the infectious processes taking place in this location [70,71]. On the one hand, ILCs can worsen the course of the disease by driving a deregulated immune response, resulting in chronic inflammation, and through IL-5 and IL-13, they promote inflammation by mediating airway hyperresponsiveness. On the other hand, ILC1s promote viral clearance, and the IFN-γ they secrete hinders viral replication [72,73,74]. Moreover, ILC2s contribute to lung regeneration in the course of viral infections, and the amphiregulin (AREG) produced by ILC2s maintains the integrity of the airway and intestinal endothelium, which, in mouse models, results in reduced severity of influenza virus infection and reduced mortality [75,76]. In their study, Gomez-Cadena et al. discovered a noteworthy decrease in the overall count of ILCs among COVID-19 patients, including both mild and severe cases. They also observed a significant rise in the ILC2 subpopulation. However, no significant changes were observed in other ILC subpopulations [77]. Silverstein et al. demonstrated a significant decrease in the number of ILCs in adult patients infected with SARS-CoV-2. Moreover, the study found an inverse correlation between the number of ILCs and the risk of hospitalization, duration of hospital stay, and severity of inflammation, as indicated by CRP (C-reactive protein) values [68] The number of ILCs during the SARS-CoV-2 infection was also markedly reduced in children and adolescents [68]. These findings were supported by Garcia et al., who conducted a study that demonstrated a decreased occurrence of ILC2s in severe cases of COVID-19. They also noted a concurrent decline in the number of ILC precursors (ILCp) in all patients analyzed, when compared to the control group. Furthermore, it was observed that ILCs undergo activation during SARS-CoV-2 infection, leading to alterations in the expression of surface proteins [77]. Activated ILC2s and ILCp showed a phenotype with an increased CD69 expression and different levels of the CXCR3 and CCR4 chemokine receptor expression [78]. In addition, ILC2s increased the NKG2D levels in patients with severe symptoms compared to patients with mild disease and controls, while differences in the NKG2D expression in ILC1s and ILCP were not observed [71]. Gomez-Cadena et al. showed in their study that in the group of ILC2 cells in patients with severe disease, the percentage of the cells expressing the NKG2D + molecule increased compared to that in patients with mild disease and the control group. It is worth mentioning that, until now, the presence of NKG2D, an activating C-type lectin-like molecule, in ILC2s—a type of cytotoxic NK cell—has not been documented. In contrast, there were no noticeable variations in the expression of NKG2D, KLRG1, or CD25 in patients with ILC1 or ILCP [77]. The levels of other markers, such as PD-1, NKG2A, and NKp46, were similar in ILC2 controls and patients with mild and severe disease [77]. There was a positive correlation observed between the levels of activated (CD69+) total ILCs and activated ILCp, as well as the levels of serum IL-6 and CXCL10 in patients with COVID-19 [71]. On the contrary, COVID-19 patients exhibited a negative correlation between the levels of CXCL10 and CXCL11 and the percentage of CXCR3+ ILCs [71]. The concept of a reduced population of ILC2s in the blood may be consistent with reports that the number of ILC2s decreases during infectious diseases under the influence of IFN-γ [76,79]. In tracheal aspirates in COVID-19, there is usually a significant increase in the levels of T cells, MAIT, and γδ, as well as ILCs (especially ILC2s) [80]. In the studies above, the number of amphiregulin-producing ILCs in the course of SARS-CoV-2 infection was found to be higher in women than in men, and those hospitalized with COVID-19 had a lower percentage of amphiregulin-producing ILCs than controls [68]. On the other hand, there are suggestions that the production of IL-17 may play a deleterious pro-inflammatory role in severe disease [71]. More comprehensive research on the involvement of ILCs in the COVID-19 infection may result in a better understanding of the disease and provide opportunities for better control and treatment in the future. This area undoubtedly requires further research.

### 3.4. Respiratory Syncytial Virus (RSV) 

Respiratory syncytial virus (RSV) is a highly contagious, single-stranded RNA virus that is responsible for widespread infections across the globe. Although RSV can cause infections throughout the year, it tends to flourish and create epidemics in temperate regions, particularly during the winter season [81]. RSV is a significant cause of respiratory tract infections, particularly in children, resulting in a range of diseases, including upper respiratory tract infections (URTI) and lower respiratory tract infections (LRTI), such as pneumonia and bronchiolitis. These infections can lead to heightened morbidity and mortality rates. Natural infection causes incomplete immunity, resulting in recurrent infections in childhood and infections in adults, including the elderly [82]. Those most at risk for severe RSV-induced illness and associated hospitalization include premature infants and infants with chronic lung disease or hemodynamically significant congenital heart disease [83]. Although a functional correlation between RSV and obstruction/asthma has not yet been proven, immunological studies indicate a shift in response toward Th2 cells, and an attenuation of the antiviral IFN-γ response during RSV infection underlies airway hyperresponsiveness in a subset of susceptible children after RSV infection [84].

When airway epithelial cells are infected with RSV, they can release thymic stromal lymphopoietin (TSLP), IL-33, HMGB1, and IL-25 upon recognition of RSV transcripts and virus replication intermediates by bronchial epithelial cells. These alarm proteins (“alarmins”) released from RSV-infected epithelial cells can activate ILC2 cells to produce the type 2 cytokines IL-4, IL-5, and IL-13. IL-5 is the most critical factor in eosinophils’ growth, differentiation, and survival. IL-13 has many immunological and physiological effects, including promoting airway reactivity and mucosal cell metaplasia [85]. When activated, ILC2s reflect the innate Th2 cell counterparts and are potent promoters of airway inflammation and hyperresponsiveness in RSV-induced bronchiolitis, as well as obstruction/asthma in children. Persistent changes in the airway epithelium’s epigenetic profile following RSV infection might contribute to a higher risk of obstructive lung disease due to RSV in young individuals, both in the short and long term [86]. ILC2s express receptors for cytokines such as TSLPR, ST2, and IL-25R, as well as receptors for advanced glycation end products (RAGE) and Toll-like receptors (TLR) 2 and 4. ILC2s and the cytokines they produce play a significant role in the development of asthma and allergic diseases. Nasal aspirates from infants hospitalized with severe RSV infection showed the presence of ILC2s, with higher levels compared to those with milder disease. Additionally, the levels of IL-33 and TSLP, which are type 2 and epithelial-derived cytokines, were significantly elevated in nasal aspirates from infants with severe disease compared to those with moderate disease. Furthermore, studies have confirmed that infants who are younger are at a higher risk of experiencing severe RSV infection. Additionally, infants with elevated levels of IL-4 in nasal aspirates also face an increased risk. These findings establish a crucial connection between ILC2 cells and the development of RSV-induced bronchiolitis in infants [85]. ILC2s constitute the main population of ILCs in the lungs and secrete large amounts of type 2 cytokines during inflammation the airways. Research indicates that ILC2s play a vital function as initial responders to RSV infection and are pivotal in coordinating the subsequent adaptive immune response. In a cohort of patients, those with moderate RSV infection displayed elevated levels of IFN-γ, IL-12p40, and IL-17A in comparison to infants with severe RSV infection. Conversely, the severe group demonstrated significantly higher levels of type 2 respiratory cytokines (IL-4 and IL-13) and IL-33 when compared to the moderate group. The frequency of ILC2s, primary type 2 immune mediators, was significantly higher in infants with severe and moderate RSV. The comparison of ILC2 levels and total cell counts in nasal aspirates revealed similar findings between infants with moderate RSV and severe RSV. Interestingly, infants older than three months who were infected with RSV had significantly lower levels of IL-4 and ILC2s in nasal aspirates compared to infants three months or younger. In contrast, the levels of IFN-γ were significantly higher in RSV-infected infants older than three months compared to infants three months or younger. Therefore, the study conducted by Vu et al. demonstrates that a higher frequency and absolute number of ILC2 in nasal aspirates are associated with more severe RSV disease, suggesting that an elevated ILC2 count increases the risk of severe RSV [87]. RSV infection induces activation of CD4+ T lymphocytes. It has been demonstrated that RSV infection leads to an increase in the expression of MHC II molecules on pulmonary ILC2s. This increase is believed to contribute to the expansion and differentiation of RSV-infected CD4+ T cells. However, when the interaction between CD4+ T cells and ILC2s was blocked using anti-MHC-II monoclonal antibodies, there was a significant reduction in CD4+ T cell expansion. These findings suggest that pulmonary ILC2s may serve as antigen-presenting cells, utilizing the MHC II pathway to activate CD4+ T cells during RSV infection. Additionally, reducing the number of CD4+ T lymphocytes may help mitigate airway inflammation caused by RSV infection [88]. 

Recent research has focused on the concept of “trained immunity” in various immune cells, such as macrophages, dendritic cells, NK cells, and ILCs. Specifically, the impact of this training on ILC2s during early childhood development and its influence on immune responses in adulthood has been explored. Studies have shown a significant increase and activation of ILC2s shortly after birth, with lineage studies indicating that a substantial portion (40–70%) of these cells persist into adulthood in the lungs. The activity of these cells is heavily influenced by their exposure during the early postnatal period. Notably, the activation of pulmonary ILC2s by IL-33 during early life appears to “train” them, resulting in their long-term survival and a stronger response to type 2 infections later in life. Furthermore, ILC2s have been found to play a critical role in shaping the immune response during RSV infections. Understanding the relationship between viral load and the severity of RSV infection is crucial for developing effective antiviral therapies, improving the management of asthma and other respiratory diseases associated with viral infections, and facilitating the validation of drugs and vaccines. Overall, elucidating the mechanisms behind trained immunity in immune cells, particularly ILC2s, has significant implications for our understanding of immune responses and the development of therapeutics for various diseases [87]. 

### 3.5. Chlamydia sp. 

*Chlamydia* sp., a member of the *Chlamydiae* phylum, is a Gram-negative, obligate intracellular pathogen, among which *Chlamydia trachomatis* (*C. trachomatis*) and *Chlamydia pneumoniae* (*C. pneumoniae*) are known as major human pathogens. C. trachomatis infection is the most common sexually transmitted disease in the world. In turn, *C. pneumoniae* causes respiratory tract infections, including atypical pneumonia, and promotes the development of chronic diseases such as asthma, arthritis, and atherosclerosis [89,90,91,92].

*Chlamydia* strains are characterized by a common biphasic life cycle, including two developmental forms: infectious elementary body form, which differentiates within the pathogen-specific inclusion into replicative reticulate body form. Their strongly reduced genome and the resulting lack of many enzymes entail developing specific interactions with the infected host. However, their small genome (~1.04 Mbp in *C. trachomatis*) and the plasmid in most chlamydia strains encode numerous virulence factors [89,93,94].

Using appropriate mouse models lacking the interleukin two receptor common gamma chain (IL-2Rγc) and the IL-7 receptor necessary for developing ILCs, Xu et al. demonstrated that ILCs are essential for endometrial innate immunity [9]. They also demonstrated that inhibition of *C. trachomatis* development depends on IFN-γ secretion. Similarly, He et al. confirmed the importance of IFN-γ in chlamydial infections, highlighting the role of ILC3s in producing this cytokine [95]. Recently, it has also been suggested that genital tract infections with *Chlamydia* spp. facilitate the plasticity of ILC3 to ILC1 [96,97].

### 3.6. Toxoplasma gondii 

*Toxoplasma gondii* (*T. gondii*) is an intracellular pathogen responsible for toxoplasmosis, a protozoan infection distributed worldwide. It is estimated that 8–22% of North Americans and 30–90% of Europeans may be infected with this parasite [98,99]. Although the course of the disease is usually asymptomatic, for immunocompromised people and pregnant women, it can pose a particular threat. *T. gondii* reproduces in cats’ small intestine, which is the definitive host, after which it is excreted as oocysts with the feces. After entering the intermediate host (birds, mammals, including humans), it multiplies in the small intestine, from where the protozoan enters various organs and tissues via the blood and lymphatic route, including skeletal muscles, the eyeball, lymph nodes, and even central nervous system tissues, where it eventually forms cysts that can persist in the body for years [100,101]. ILC1 cells are one of the keys to immunity against *T. gondii* infection. Activated ILC1 cells play a vital role in initiating an immune response by producing pro-inflammatory cytokines, such as IFN-γ and TNF-α [102]. To effectively combat intracellular pathogens, a strong innate immune response called type I is required, which involves the production of IFN-γ. This essential cytokine is necessary for fighting against bacterial, viral, and parasitic infections. While the specific mechanism by which IFN-γ produced by these innate immune cells defends against *T. gondii* remains unknown, López-Yglesias et al. demonstrated that the early production of IFN-γ by ILC1s and NK cells, regulated by T-bet, is critical for the survival of DCs during infection. Furthermore, T-bet-controlled innate IFN-γ is crucial for inducing the transcription factor IRF8, which is vital for maintaining inflammatory DCs. They demonstrated that IRF8+ DCs are essential for eliminating the parasite [103]. Snyder et al. demonstrated that the absence of MyD88 in a mouse model affects the function of ILCs, depending on the location of *T. gondii* infection. Furthermore, they found that this effect is mediated by a mechanism that is dependent on MyD88. In mice lacking MyD88 and infected orally, the frequencies of T-bet+ ILC1s producing IFN-γ in the small intestine were lower compared to wild-type mice. When MyD88 knockout mice were treated with antibiotics to deplete microflora, the frequencies of IFN-γ-producing ILC1s were further reduced. During intraperitoneal (i.p.) infection in mice, the peritoneal cavity primarily exhibited polarization of ILC towards the ILC1 subset, with enhanced expression of IFN-γ observed throughout the infection. This response was driven by IL-12p40 and associated with ILC proliferation. In MyD88-/- i.p. infected mice, IFN-γ expression by ILC1 was not maintained, but proliferation remained normal [104]. It has been observed that ILC1s are the early producers of IFN-γ and TNF in response to cerebral *T. gondii* infection. This activation of host defense mechanisms and the resulting neuroinflammatory response are triggered by the ILC1s. In a mouse model, it was demonstrated that during the initial stages of cerebral *T. gondii* infection, ILCs accumulate in the cerebral parenchyma, choroid plexus, and meninges. Furthermore, when antibodies are used to deplete both NK cells and ILC1s early on in the infection, the expression of cytokines and chemokines decreases, while the number of parasites in the brain increases. It suggests that the absence of ILC1s explicitly affects immune responses in the brain [105].

During *T. gondii* infection, NK cells and ILC1 activation are observed, manifested by the production of IFN-γ and TNF-α. IL-12, released by dendritic cells in response to the rapidly replicating tachyzoites of the parasite (acute inflammation), participates in activating these cells. This is further followed by the formation of the parasite’s slow-replicating developmental stage, bradyzoites, which are encysted in host cells (the chronic form). It was reported that there was an expansion of the cells resembling ILC1 in the spleens of *T. gondii*-infected mice, which were named ILC1-like cells, while there were small populations of ILC1-like spleen cells in uninfected mice. Interestingly, although T. gondii infection induces the formation of these cells, it appears that they can persist independently of ongoing parasite replication, suggesting a permanent change [105]. There is some evidence that although ILC1-like cells differentiate from NK cells, they are distinct from NK cells and ILC1. It is suggested that the enduring presence of ILC1-like cells, even after the infection has been eliminated, resembles the traditional concept of immune memory, implying that ex-NK cells have a broader impact in *T. gondii* infection than what has been previously associated with NK cells and ILC1s [106].

Very little is known about whether ILC2s play a role in *T. gondii* infection. These cells are described primarily as part of the host defense response against helminths (*Nippostrongylus brasiliensis* [107,108], *Heligmosomoides polygyrus* [109], *Strongyloides venezuelensis* [110], *Trichinella spiralis* [111], *Ttichuris muris* [112], *Schistosoma haematobium* [113]); however, they may affect the anti-*T. gondii* response. In infected mice, there was a noticeable thinning of the epidermis and an infiltration of inflammation in the skin, accompanied by a reduction in the quantity of mast cells in comparison to mice that were not infected. The findings correlated with the decrease in allergic responses of both Th2 and Th1 types. During the initial 24-h period of allergic sensitization, the levels of type 2 cytokines IL-4 and IL-5 were found to be diminished in the splenocytes and draining lymph nodes of infected mice. Moreover, the decreased type 2 profile observed in animals with chronic infection was associated with a decline in the ILC2 presence within the draining lymph nodes.

Little is known about the role ILC3 plays in *T. gondii* infection. Since *T. gondii* is an orally acquired pathogen, and ILC3s are present in tissues, including the gut and mucosal barriers, these cells may also be involved in the course of infection with this pathogen. Special attention is paid to the two central cytokines produced by ILC3s, IL-22 and IL-17, which determine the maintenance of homeostasis and integrity of mucosal barriers. It is believed that, while the first one helps restore and protect the intestinal mucosal barrier, the latter can promote inflammation and subsequent pathological changes [114]. However, the role of IL-17 is not unequivocally protective or destructive. Many reports point to IL-17 as a double-edged sword in host immunity to certain infections [115]. Although, in the case of *T. gondii* infection, it has been shown that IL-17 can induce intestinal immunopathological changes and promote the development of the chronic form of the infection, there are minimal reports identifying ILC3s as the source of this cytokine [116]. On the contrary, ILC3s can produce IL-17A and IL-17F when stimulated by IL-1β and IL-23, leading to the production of chemokines that attract neutrophils from epithelial cells, including those in the gut. Macrophages can also be activated by IL-17 and IFN-γ and work alongside neutrophils to eliminate intracellular bacteria, fungi, and protozoan parasites through phagocytosis and killing. Additionally, IL-17A, IL-17F, and IL-22 can boost the production of antimicrobial peptides (AMPs) and enhance the functioning of the epithelial barrier [115]. Understanding the function of ILC cells during *T. gondii* infection is crucial, as it could contribute to developing new treatments and vaccines. By manipulating the activity of ILC cells, it may be possible to enhance parasite eradication and increase anti-*T. gondii* therapy effectiveness.

### 3.7. Streptococcus pneumoniae 

*Streptococcus pneumoniae* (*S. pneumoniae*) is a Gram-positive, facultative anaerobic bacterium that colonizes the nasopharynx and causes community-acquired pneumonia [6]. The severity of the disease depends on the strength of the inflammatory response triggered by the activation of complement pathways and the release of cytokines stimulated by bacterial cell wall proteins, envelope polysaccharides, and DNA [117]. The polysaccharide capsule plays a significant role in the pathogenesis of *S. pneumoniae* infection, which interferes with phagocytosis by inhibiting the binding of the complement component C3b to the cell surface. Additional factors that contribute to the virulence of the bacteria include pneumolysin, pneumococcal surface protein A, autolysin, and pili. The significance of ILC3s in *S. pneumoniae* infections was demonstrated by van Maele et al. and Gray et al. [6,118]. Through the use of mouse lung infection models, van Maele et al. discovered that intranasal *S. pneumoniae* infection resulted in an increase in the levels of IL-22, IL-17A, and IL-17F in the lungs. ILC3s, which expressed retinoic acid-related orphan receptor γt (RORγt) and chemoattractant cytokine (chemokine) C-C receptor 6 (CCR6), were the only cells that co-expressed these markers, while the NK or NKT cells did not. These ILC3s were also found to strongly upregulate the expression of chemokine (C-C motif) ligand 20 (CCL20) in lung tissues during infection. Moreover, the administration of flagellin, a Toll-like receptor five agonist, enhanced IL-22 and IL-17 production by ILC3s and reduced the bacterial load in mice [6]. Gray et al. revealed that 90% of IL-22-producing cells in the lungs of newborn mice had phenotypic markers of ILC3s. The deficiency of ILC3s increased the susceptibility of newborn RORγtiDTR mice (with depletion of the RORγt+ T cells after diphtheria toxin (DT) treatment) to *S. pneumoniae*. By adopting the transfer of ILC3s into mice, their immunity to *S. pneumoniae* was successfully restored. Additionally, it was discovered that administering IL-7 intranasally, which is a crucial factor for RORγt+ cell function, led to an augmentation in the quantity of innate RORγt+ T cells in the lungs. This, in turn, resulted in an enhancement of IL-17A expression and a reduction in the bacterial burden following *S. pneumoniae* infection [119]. A study by Saluzzo et al. showed that lung-resident ILC2s significantly contribute to the phenotype and function of resident alveolar macrophages. IL-33 derived from ILC2s, which predominate in the lung environment in the early postnatal period, may be detrimental in *S. pneumoniae* infection due to the induction of the development of M2-type alveolar macrophages [120]. 

### 3.8. Klebsiella pneumoniae 

*Klebsiella pneumoniae* (*K. pneumoniae*) is a Gram-negative, aerobic, rod-shaped bacterium that can cause urinary, respiratory, gastrointestinal, and skin infections [121]. A wide range of virulence factors, together with its common highly antibiotic-resistant phenotype, makes the pathogen a serious health threat to neonates, the elderly, and immunocompromised patients. Innate immune defense against *K. pneumoniae* has been shown to depend on the cooperation between ILCs and inflammatory monocytes [122]. Xiong et al. conducted a study that showed how inflammation-causing monocytes recruited to the lungs of *K. pneumoniae*-infected mice produced TNF. This led to a notable increase in the occurrence of ILCs that produced IL-17A. This cytokine played a role in enhancing the ability of monocytes to take up and eliminate bacteria, suggesting that there is a reciprocal relationship between innate lymphocytes and monocytes that aids in the resolution of lung inflammation [122]. Other studies have also proven the importance of IL-17 in the pulmonary host defense against *K. pneumoniae* [123]. Using a mouse model of *K. pneumoniae* infection, Chen et al. found that, 24 h after infection, IL-17R knockout mice had a significantly higher bacterial load compared to the wild-type animals [123]. All IL-17R knockout mice became infected within 48 h, whereas only 20% of the wild-type mice became infected within the same period. Additionally, it was shown that the production of IL-17A, which is dependent on IL-23, is crucial for the survival of adult mice when challenged with *K. pneumoniae* [124]. In mice deficient in T and B cells (Rag2-/- mice), the main source of IL-17 is ILC3s [122]. When these cells were depleted using anti-CD90 antibodies, the expression of IL-17 was eliminated, and the pulmonary infection caused by *K. pneumoniae* worsened. Through single-cell RNA sequencing, Iwanaga et al. discovered that ILC3 cells expressing IL-17, IL-22, and inducible T cell costimulatory molecule (ICOS) were necessary for protection against carbapenem-resistant *K. pneumoniae* [125]. 

### 3.9. Pseudomonas aeruginosa 

*Pseudomonas aeruginosa* (*P. aeruginosa*) is a rod-shaped bacterium that is Gram-negative and heterotrophic. It is known for causing chronic infections in patients who are severely ill or have compromised immune systems [126,127]. The main risk groups are patients with cystic fibrosis, cancer, acquired immunodeficiency syndrome (AIDS), burns, and non-healing diabetic wounds, as well as those on mechanical ventilation [126]. The pathogen uses various virulence factors for its adhesion and colonization, including suppression of host immunity and escape from immune system mechanisms. *P. aeruginosa* cells are equipped with lipopolysaccharide, flagella, pili, numerous secretion systems, proteases, and toxins and have the ability for quorum-sensing and biofilm formation [128]. *P. aeruginosa* also exploits several drug-resistance mechanisms that make the infection difficult to eradicate [128]. Using two knockout (KO) mouse strains, the RAG KO (lacking the ‘classical’ T cells αβ and γδ TCR) and the double RAG γC KO (lacking T cells, NK cells, and ILCs), Villeret et al. found that both ILCs and TCR-bearing T cells were necessary for protection against *P. aeruginosa* [127]. Compared to the wild-type animals, reduced production of IL-17 and IL-22 was observed in the lungs of *P. aeruginosa*-infected RAG KO and double RAG γc chain KO mice. There was also a noticeable increase in the *P. aeruginosa* burden in the lungs of double RAG γC KO mice, indicating a potential role of ILCs in controlling the infection [127].. The importance of IL-17 in clearing *P. aeruginosa*-induced lung infections was confirmed in a mouse model of lung infection using agar beads loaded with *P. aeruginosa* [129]. The study results indicated that mice with a knockout of IL-17R exhibited a higher bacterial load and experienced weight loss 14 days post-infection when compared to the control group. Furthermore, the production of IL-17 seemed to be dependent on ILC3s, as approximately 90% of CD3+ cells in the lungs that produced IL-17 displayed phenotypic markers associated with ILC3s [129].

### 3.10. Bordetella pertussis

*Bordetella pertussis* (*B. pertussis*) is a Gram-negative aerobic bacterium that is the causative agent of a severe respiratory disease called whooping cough [130,131]. The set of B. pertussis virulence factors enabling the bacteria to invade and persist within the host includes adhesins that facilitate attachment to target host cells and toxins that allow the pathogens to evade the host immune system. These factors primarily include filamentous haemagglutinin (FHA), serotype-specific fimbriae, pertactin, pertussis toxin (PT), adenylate cyclase toxin, dermonecrotic toxin, tracheal cytotoxin, lipopolysaccharide (LPS), and tracheal colonization factor (Tcf) [132]. In a mouse model of *B. pertussis* infection, Byrne et al. showed that NK cells contribute to antibacterial immunity by activating IL-12-mediated IFN-γ production, which increases macrophage activity and promotes Th1 cell differentiation [133]. Furthermore, the study revealed that the activation of NK cells by *B. pertussis* was dependent on the activation of the NLRP3 inflammasome in macrophages. This activation triggered the release of IL-18 and IL-1β through caspase-mediated mechanisms, leading to an intensified proinflammatory response against the pathogen [134]. Another study in a mouse model of *B. pertussis* infection showed inhibition of IL-22-secreting ILC3s activity by IL-23-secreting dendritic cells, hypothesizing that the pathogen could disrupt the IL-23/IL-22 axis pathway due to PTX production [130].

### 3.11. Clostridium difficile

*Clostridium difficile* (*C. difficile*) is a Gram-positive spore-forming, toxin-producing anaerobe [135]. The pathogen causes infection following disruption of the local microflora, usually due to previous antibiotic treatment [135]. Antibiotic treatment of *C. difficile* infection is often ineffective, and complications associated with recurrent *C. difficile* infections in hospitalized patients interfere with medical therapies until the infection is controlled [136,137]. The virulence of *C. difficile* is due to the presence of toxin A (TcdA) and toxin B (TcdB), both of which contribute to the glucosylation of small GTPases, including Rho, Rac, and Cdc42. As a result, the severity of the disease ranges from mild diarrhea to severe inflammatory complications, such as pseudomembranous colitis, sepsis, and even death [138]. Some experimental studies revealed that ILCs were essential to the recovery from *C. difficile* infection [139,140]. A study by Abt et al. using the C57BL/6, Rag1-/- mice lacking T- and B-cells showed upregulation of ILC1 and ILC3-related proteins, such as those derived from ILC1s IFN-γ, TNF-α, and nitric oxide synthase 2 (NOS2), as well as proteins derived from ILC3s IL-22, IL-17a, and regenerating islet-derived protein three gamma (RegIIIγ), following *C. difficile* infection. In contrast, ILC-deficient Ragγc-/- mice showed an increased susceptibility to *C. difficile* infection, which could be restored by adoptive ILCs transfer. The loss of ILC1s expressing IFN-γ or T-bet in Rag1-/- mice was observed to be linked to enhanced vulnerability to infection, highlighting the protective function of ILC1s in the immune response against *C. difficile* [139]. 

### 3.12. Helicobacter pylori

*Helicobacter pylori* (*H. pylori*) is a Gram-negative, microaerophilic bacterium that can change its form from a spiral to a coccoid, colonizing the human gastric mucosa [141]. Chronic *H. pylori* infection is linked to the development of gastric cancer and ulcers. While urease, outer membrane proteins, and flagella are among the virulence factors of *H. pylori* that contribute to bacterial colonization, the development of disease and immune escape are primarily mediated by cytotoxin-associated gene A (CagA), cytotoxin-associated gene pathogenicity island (cagPAI), and vacuolating cytotoxin type A (VacA) [142]. Many studies have shown that ILCs activity contributes to maintaining intestinal homeostasis and resistance to gastrointestinal pathogens [141,143,144,145]. Li et al. and Satoh-Takayama et al. demonstrated that ILC2 is the predominant innate immune cell subset in the stomach of *H. pylori*-infected humans and mice, whose activation can be stimulated by local commensal microflora [144,145]. Triggered by functional IL-5-producing ILC2, IgA production by B cells inhibited *H. pylori* growth and provided protection against infection [145,146]. A recent study of asymptomatic patients infected with *H. pylori*, using single-cell RNA sequencing (scRNA-Seq) and flow cytometry, showed increased numbers of NKp44+ ILC3s, as well as CD11c+ myeloid cells and activated CD4+ T cells and B cells in the gastric mucosa [147]. 

### 3.13. Candida albicans

The fungus *Candida albicans* (*C. albicans*) is a common cause of candidiasis, an infection of the skin, mouth, esophagus, gastrointestinal tract, vagina, and vascular system, particularly in immunocompromised individuals. The infection can cause mucocutaneous disorders or a potentially fatal invasive disease that affects several organs and systems [148,149,150,151]. The pathogenicity of *C. albicans* is based on the production of proteins necessary for adhesion and invasion, including adhesins that recognize fibrinogen or fibronectin, e.g., agglutinin-like sequence 3 (Als3) or hyphal wall protein 1 (Hwp1), and hydrolytic enzymes that facilitate invasion of host tissues, such as secreted aspartyl protease (SAP), phospholipase and hemolysin [149]. *C. albicans’* pathogenic properties include biofilm formation and phenotypic switching [149]. Evidence that *C. albicans* infection and colonization is under control of IL-17-secreting ILCs comes from a mouse model of oropharyngeal candidiasis [152,153]. Gladiator et al. discovered that mice lacking expression of recombination activating gene 1 (RAG1), which results in ILCs, were incapable of controlling mucosal *C. albicans* infection. Additionally, they observed a similar outcome in mice with a deficiency in retinoic acid-related orphan receptor (Rorc-/-), which also leads to ILC deficiency [152]. It was also shown that IL-17A and IL-17F, which are crucial for pathogen clearance, were produced immediately after infection in an IL-23-dependent manner by the ILCs present in the oral mucosa [152]. However, another study by Conti et al. reported that IL-17 was mainly expressed by natural T helper 17 (nTh17) and γδ T cells, but not by ILCs [154]. Subsequently, Sparber et al. showed that tongues infected with *C. albicans* had three distinct IL-17-producing cell types: nTh17s, γδ T cells, and ILC3s [153]. While deletion of all three subsets is reminiscent of the high vulnerability of IL-17RA or IL-17RC-deficient mice to *C. albicans*, the lack of nTh17 or γδ T cells does not affect fungal control [150,153].

## 4. Therapeutic Potential of ILCs in the Battle against Infections

With increasing data on the biology of ILCs and their involvement in the host immune response, they have been widely proposed as therapeutic targets in various diseases [13,151,155,156]. Currently, available therapeutic strategies targeting ILCs include cytokine delivery, adoptive transfer, anti-cytokine antibodies, antibody deprivation of ILCs, modulation of ILC plasticity and/or function, inhibition of ILC migration and function, modulation of immune checkpoints, as well as the use of lipid mediators, glucocorticoids, or beta-2 adrenergic receptor (β2AR) agonists [157].

The plasticity of ILCs and their ability to reversibly differentiate from one type to another may offer an opportunity to restore tissue homeostasis disrupted by infection. Bernink et al. found that CD14+ dendritic cells, observed in higher percentages in patients with Crohn’s disease, promoted the polarization of IL-22-producing ILC3 cells into IFN-γ-producing CD127+ ILC1 cells. On the contrary, CD14-dendritic cells promoted the differentiation of CD127+ ILC1 towards ILC3, providing hope for therapeutic modification of the composition, function, and phenotype of ILCs in the gut [13]. Buonocore et al. showed that neutralizing IL-17 and IFN-γ, the effector cytokines of ILC3s, could be therapeutically effective in bacteria-induced colitis [158]. Additional data have provided stronger evidence supporting the notion that targeting cytokines like IL-12 or IL-23, which activate ILCs to produce IL-17 and IFN-γ, could be a superior approach in the treatment of Crohn’s disease [159]. In particular, ustekinumab, a monoclonal antibody that inhibits IL-12/23 p40, has demonstrated the ability to modulate the composition of ILC subsets in the intestines of patients with Crohn’s disease, thereby leading to improved treatment outcomes [160]. In addition, Chevalier et al. noted that blockage of bacterial type 1 fimbrin D-mannose specific adhesin (FimH) with TAK-018 inhibited the adhesion of bacteria to the intestinal epithelium, preventing the mucosal inflammation associated with Crohn’s disease [161]. Currently, Sibofimloc, a gut-restricted small molecule FimH-blocker designed to treat the underlying cause of Crohn’s disease and maintain patients in a non-inflammatory disease state, is undergoing phase 2 clinical trials and is being developed under a global license [162].

Janus kinase inhibitors affecting intracellular signaling are also considered promising candidates for targeting ILCs. A monoclonal antibody against the natural killer group 2D (NKG2D) receptor, a molecule that is constitutively expressed on human NK cells, induced clinical remission of Crohn’s disease, most likely as a result of blocking lymphocyte cytotoxicity and production of cytokines [163]. The use of fingolimod (FTY720) or SEW2871, the antagonists of sphingosine-1 phosphate receptor 1 (S1PR1), which modulates ILC migration from secondary lymphoid organs, was also found to regulate the production of cytokines (GM-CSF, IL-22, IL-17, and IFN-γ) by ILC1s and ILC3s, thereby reducing tissue inflammation [164].

Several studies have revealed that mutations in some genes regulating the functions of ILCs, such as those encoding Janus kinase 3 (JAK3), interleukin two receptor gamma (IL2RG), or integrin subunit beta 2 (ITGB2), can lead to reduced numbers of ILCs or their impaired activity [165]. Patients with such genetic mutations have been shown to suffer from a profound defect in NK cell development and exhibit an increased susceptibility to various infections [166]. In vitro and in vivo studies have revealed that NK cells are active against a wide range of viral, bacterial, and fungal pathogens, suggesting that they may be a promising tool for antimicrobial treatment. Most current research focuses on the adoptive transfers of NK cells to a host suffering from infectious complications and genetic modifications of NK cells and their receptors [151]. Parker et al. demonstrated that transferring activated NK cells into *Aspergillus fumigatus*-infected mice led to improved elimination of the pathogen from the lungs of both IFN-γ-deficient and wild-type animals [167]. The use of antibodies or inhibitors of CD27 signaling might represent another potential therapeutic intervention. Induced overproduction of IFN-γ by CD27+ NK cells was observed in mice infected with *Listeria monocytogenes*, resulting in decreased levels of CXCR2 on granulocytes and inhibition of granulocyte recruitment at the site of infection [168]. However, mice can be rescued from lethal Listeria monocytogenes infection by depleting IFN-γ or blocking CD27 signaling with specific antibodies [168]. Similarly, a reduction in the number of NK cells during *Pseudomonas aeruginosa* infection was accompanied by faster clearance of the bacteria and better prognostic parameters [169]. Hence, additional research is required to comprehensively determine the various factors through which the adoptive transfer of NK cells can either benefit or harm the host.

## 5. Concluding Remarks

Over the past ten years, ILCs have been recognized as an essential immune response component for maintaining barrier resistance against infectious agents. Due to their ability to rapidly secrete immunoregulatory cytokines, they are central to the innate immune response, playing an essential role in shaping the adaptive response through interaction with other immune cells. The microenvironment of the tissue in which they reside determines their functional diversity, enabling them to act in multiple effector functions. However, many unanswered questions still make it difficult to fully understand the complex function these cells can play in both health and disease. Future studies should further investigate the modulation of ILC activity by various infectious agents in animal and human models to increase the translation of basic research findings into developing new effective antimicrobial therapies targeting ILCs.

## Figures and Tables

**Figure 1 cells-13-00335-f001:**
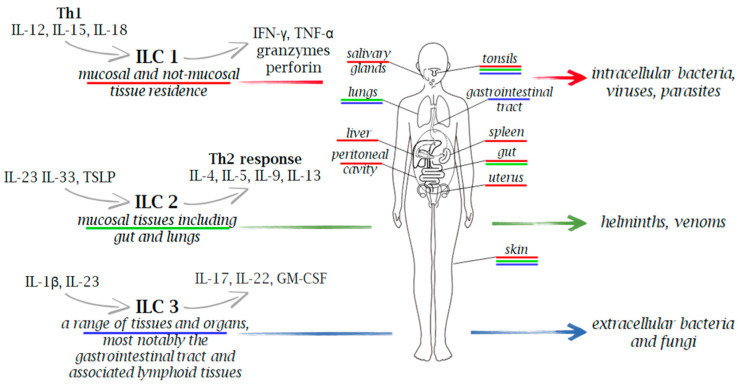
Innate immune cells in immunity to extracellular and intracellular pathogens. Due to the wide distribution of heterogenous subtypes of ILCs across various tissues and organs in the body, these cells play a crucial role in the immune response against a range of pathogens, including viruses, bacteria, fungi, and both intracellular and extracellular parasites. Upon stimulation, ILCs secrete a variety of cytokines, with IFN-γ for ILC1, IL-5 and IL-13 for ILC2, and IL-17 and IL-22 for ILC3, serving as their signature cytokines.

**Figure 2 cells-13-00335-f002:**
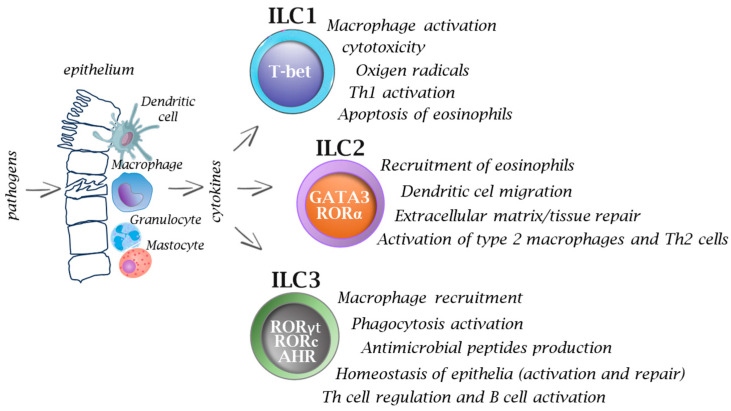
Functions of innate immune cells in infections. After pathogen invasion, the interplay between innate immune cells and ILCs provides signals (e.g., cytokines) that activate ILCs, and thus they play an important role on in the immune response from the very beginning, while T cells, based on their receptor specificity, must undergo a process of selection and further multiplication, which usually takes several days. Activation of ILCs occurs *via* specific transcription factors which ultimately allows the cells to participate in many immune processes. Inappropriate or prolonged activation of ILCs can lead to excessive inflammation and tissue damage.

## Data Availability

The data presented in this study are openly available under reference numbers.

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
