# Peer review of "Innate Lymphoid Cells and Their Role in the Immune Response to Infections"

_cells, 2024, doi:10.3390/cells13040335_

Round 1
Reviewer 1 Report
Comments and Suggestions for Authors
The present manuscript provides a comprehensive overview of ILC functions in a broad range of infections. It is well-organized and clear, making it accessible even for novice researchers in the field.
Author Response
The authors thank the Reviewer for appreciating the manuscript.
Reviewer 2 Report
Comments and Suggestions for Authors
The manuscript entitled “Innate lymphoid cells and their role in the immune response to 2 infections” summarizes the functions of ILCs in 24 the context of the immunology of infections caused by different intracellular and extracellular pathogens and discusses their possible therapeutic potential”.The manuscript needs minor revision before it can be accepted for publication.
1. The author should improve the english grammer and make the sentences grammatically correct.
2. The authors need to add a few more recent references relevant to the work.
3. Concluding remarks heading is written with red color while all other headings are with black color, in order to bring uniformity please change the heading color.
4. It will be great if the authors can change the color of figure legends from red to some other color like blue or black.
Comments on the Quality of English LanguageSome minor grammatical changes need to be done.
Author Response
- The author should improve the english grammer and make the sentences grammatically correct.
The grammar of the text has been improved.
- The authors need to add a few more recent references relevant to the work.
The references have been added.
- Concluding remarks heading is written with red color while all other headings are with black color, in order to bring uniformity please change the heading color.
The colour of the letters has been changed.
- It will be great if the authors can change the color of figure legends from red to some other color like blue or black.
The colour of the legend has been changed.
Reviewer 3 Report
Comments and Suggestions for Authors
This review is focused on the function of ILCs in infections caused by different intracellular and extracellular pathogens. The manuscript is well-written and organized. However, there are some minor points to be modified and suggestions for improving the text:
1) Authors could add some information about a recently identified subpopulation of ILCs, named regulatory ILCs (ILCregs), that share immunosuppressive features with Tregs and suppress the activation of other ILCs.
2) Authors should provide an in-text citation for any figures that they reproduce in their work. Authors did not refer to Figures 1 and 2 in the text.
Author Response
1) Authors could add some information about a recently identified subpopulation of ILCs, named regulatory ILCs (ILCregs), that share immunosuppressive features with Tregs and suppress the activation of other ILCs.
The information on ILCregs has been added (page 1-2)
2) Authors should provide an in-text citation for any figures that they reproduce in their work. Authors did not refer to Figures 1 and 2 in the text.
The figures were created by the authors of the paper. Both figures were quoted in the paper.